# Factors that influence the feasibility and implementation of a complex intervention to improve the treatment of peripheral arterial disease in primary and secondary care: a qualitative exploration of patient and provider perspectives

Clair Le Boutillier [1,2] Athanasios Saratzis,[3] Prakash Saha,[4] Ruth Benson,[5] Bernadeta Bridgwood,[6] Emma Watson,[3,6] Vanessa Lawrence [1]

**Correspondence to**
Dr Vanessa Lawrence;
vanessa.c.lawrence@kcl.ac.uk and
Dr Clair Le Boutillier;
clair.le_boutillier@kcl.ac.uk

## ABSTRACT

**Objectives** Our aim was to examine the feasibility and implementation of a complex intervention to improve the care of patients with peripheral arterial disease (the LEGS intervention) from the perspective of patients, general practitioners and secondary care clinicians.

**Design** A qualitative study involving semistructured individual interviews with patients and providers to gain an understanding of the feasibility of the LEGS intervention as well the barriers and facilitators to implementation in secondary and primary care.

**Setting** Primary and secondary care settings across two National Health Service Trusts.

**Participants** Twenty-five semistructured telephone interviews were conducted with (1) patients who had received the intervention (n=11), (2) secondary care clinicians responsible for delivering the intervention (n=8) and (3) general practitioners (n=6).

**Analysis** Data were initially analysed using inductive descriptive thematic analysis. The consolidated framework for implementation research was then used as a matrix to explore patterns in the data and to map connections between the three participant groups. Lastly, interpretive analysis allowed for refining, and a final coding frame was developed.

**Results** Four overarching themes were identified: (1) the potential to make a difference, (2) a solution to address the gap in no man's land, (3) prioritising and making it happen and (4) personalised information and supportive conversations for taking on the advice. The impetus for prioritising and delivering the intervention was further driven by its flexibility and adaptability to be tailored to the individual and to the environment.

**Conclusions** The LEGS intervention can be tailored for use at early and late stages of peripheral arterial disease, provides an opportunity to meet patient needs and can be used to promote shared working across the primary–secondary care interface.

## STRENGTHS AND LIMITATIONS OF THIS STUDY

⇒ A strength of this study is the focus on rigour and triangulation of participant perspectives.

⇒ We used implementation theory to guide analysis and to provide a layered coding approach, to ensure that intervention and implementation recommendations are novel, grounded in the data and useful.

⇒ While the anonymity of telephone use can allow participants to disclose sensitive information, telephone interviews have received criticism for compromising interviewer/participant rapport and interaction, and for limiting contextual data due to the absence of face-to-face contact and visual cues. However, this method of data collection is convenient, in that it is flexible (in terms of time and location), accessible (ie, remote research conducted during the COVID-19 pandemic), and allows for a wide reach (eg, accessing a diverse population from both urban and rural areas).

⇒ Although the sample size for this study may seem small, the volume of interview data we collected is unique for this patient population. Living with the symptoms of peripheral arterial disease is challenging, and patients also experience barriers to accessing healthcare. Engaging in research alongside managing these specific healthcare and health needs can be burdensome.

## BACKGROUND

Peripheral arterial disease (PAD) affects a fifth of people over the age of 60 in the UK. This prevalence is also expected to rise due to sedentary lifestyles, poor dietary choices, a longer life-expectancy and rising numbers of those with diabetes.[1 2] Patients with PAD who develop symptoms present either with

intermittent claudication (IC) or in some cases, chronic limb threatening ischaemia (CLTI) where the reduction in blood flow is so severe that it causes pain at rest, ulceration or gangrene.[3] Patients with IC are more likely to suffer a cardiovascular event compared with age-matched individuals, and the 1 year risk of limb amputation for those with CLTI is 30%. More than half of those diagnosed with symptomatic PAD are expected to die, have an amputation or a major cardiovascular event within 5 years.[4–7]

The treatment of these high-risk individuals is shared between primary and secondary care. While established national and international guidance[8] outlines how PAD risk factors should be addressed to prevent cardiac events and amputations, a study conducted by our group found that only a tenth of those diagnosed with PAD were prescribed appropriate preventive medication, and almost none received structured lifestyle or dietary support.[9] As a result, and following patient and public engagement (ie, interviews and workshops with patients and a national consultation with primary and secondary care clinicians) and a review of the existing evidence, a complex intervention in the form of a care bundle was developed to support patients, general practitioners (GPs) and secondary care clinicians to actively manage cardiovascular risk factors, and to improve implementation of existing guidance when treating patients with PAD. The care bundle, called the LEGS intervention (Leaflet, Gp letter, Structured checklist) has three elements: (1) a one-page clinical checklist for the ward and a separate one-page clinical checklist for the secondary care clinic, (2) a patient and carer education leaflet (developed with patients) and (3) a GP action letter, which can be sent to GPs at regular intervals to inform them of the PAD diagnosis and corresponding practice guidance. The LEGS intervention and full study protocol are available online (isrctn.com/ISRCTN13202085).

This qualitative study was embedded within the Community and Hospital cAre Bundle to improve the medical treatment of cLaudIcation and critical limb iSchaemia (CHABLIS) study. CHABLIS is a multicentre prospective mixed-methods cohort study that aims to determine the feasibility and effectiveness of the LEGS intervention. The aim of the qualitative study was to gain an understanding of the feasibility and implementation of the LEGS intervention, by exploring patient and provider perspectives on receiving and delivering the intervention.[10] Guided by the MRC framework for developing and evaluating complex interventions, the objectives of the qualitative study were as follows: (1) to explore patient perspectives on receiving the LEGS intervention, (2) to gather secondary care clinician views on feasibility and to understand factors that influence the success of implementation, and (3) to identify barriers and facilitators that influence the success of future implementation from the GP perspective.

## METHODS
### Study design
The study used qualitative methods to provide insights into the feasibility and acceptability of the intervention as well as the complexities of implementation.[10] Semistructured individual interviews were conducted with patients and providers to gain an understanding of the barriers and facilitators to implementation in secondary and primary care. The Consolidated criteria for Reporting Qualitative research guideline was used to ensure quality when reporting the research.[11]

To meet objective 1, semistructured individual interviews were conducted with patients with symptomatic PAD (ie, IC, ischaemic rest pain or CLTI), who had received the intervention (either in an outpatient clinic or as a part of inpatient treatment). To meet objective 2, semistructured individual interviews were conducted with secondary care clinicians who were responsible for delivering (ie, delivering or intending to deliver) the intervention, including junior doctors, vascular nurses and vascular surgeons. Staff and patients were recruited from two implementation sites: University Hospitals of Leicester National Health Service (NHS) Trust (UHL) and Guy's and St Thomas' NHS Foundation Trust. To meet objective 3, semistructured individual interviews were conducted with GPs to gain an understanding of the barriers and facilitators to future implementation in primary care and across the NHS. GPs were not involved in delivering the study intervention and were asked to report on perceived factors that might influence implementation beyond the context of the study.

Participants were selected purposively (both those who were receiving or responsible for delivering the LEGS intervention) on the basis that they are the target adopters of behaviours that influence the success of future implementation and/or could offer a particular perspective on implementation. In total, 106 patients received the intervention. All patients who had received the intervention and who gave consent for the researcher to make contact about the qualitative study were approached for participation (n=26). All secondary care clinicians responsible for delivering the intervention (n=13) were invited to take part. Secondary care clinicians were invited to participate following an in-person meeting with the local principal investigator at each site. A participant information sheet that outlined the study was provided to all potential participants with time to consider whether or not they would like to take part. GP participants were recruited from Clinical Commissioning Groups linked to the two secondary care implementation sites, chosen to enhance joined up working and to provide a local perspective on implementation across community and hospital settings. Potential GP participants (n=7) were first approached by the local Clinical Research Network research delivery manager or local contacts, and subsequently, recruited by the lead author via telephone or email. Written or verbal informed consent was obtained from each participant after a full explanation and information leaflet was given

and time allowed for consideration. Participants were not reimbursed for participating in the research. The right of each participant to refuse to participate without giving reasons was respected. All participants were free to withdraw from the study at any time without giving reasons and without prejudicing further treatment or employment.

## Data collection

Interviews used flexible open-ended questions for early data collection to gather a rich and detailed understanding of participants' perspectives (included in online supplemental file 1). The patient and provider interview schedules were revised iteratively in response to the priorities and concerns of participants and gathered information on factors that facilitate or hinder implementation and on possible solutions to overcome any identified barriers. Interviews were conducted by telephone by the lead author (a qualitative research fellow), lasted around 45 min, were audiorecorded and transcribed verbatim. Researcher reflexive notes were kept after each interview to consider the interaction with the participant, and to detail initial thoughts. Interviews were conducted between September and November 2021.

## Data analysis

Data from the three participant groups (patients, secondary care clinicians and GPs) were first analysed separately using inductive thematic analysis, where analytical concepts and perspectives are generated from the data in a deliberate and systematic way.[12] Data analysis began with repeated rereading of individual transcripts and relistening of sound files for data immersion. This was followed by line-by-line open coding, where extracts were coded under one or several descriptive themes, that is organising the data according to semantic content, to capture their meaning and reflect the content of the data. Each theme was refined, and where data allowed, further subthemes were developed. The Consolidated Framework for Implementation Research (CFIR) was then used as a matrix to organise the early themes, to explore patterns and relationships in the data, and to map connections between the themes and the three participant groups.[13] Lastly, interpretive analysis allowed for the refining of the specifics of themes and thematic patterns, and a final coding frame for patients and providers was developed. Refinements to the specifics of themes, and thematic patterns continued until a useful and meaningful analysis was achieved.[14]

Data collection occurred concurrently with data analysis; NVivo QSR International qualitative analysis software (V.12) was used to manage the data.[15] The lead author (CL) directed and conducted the analysis. Coding by a second analyst (VL) was undertaken to provide an opportunity to reflect on the coding approach, and to enhance the interpretive depth of the data.[14] This involved an independent review of the data, the themes and the CFIR mapping/tabulation. The overarching coding framework was developed following discussion with both analysts.

## Patient and public involvement

The LEGS intervention and CHABLIS mixed-methods study design were developed in collaboration with a stakeholder PPI group that included patients, family members and carers, and clinicians working in primary and secondary care.[16]

## RESULTS

## Participants

A total of 25 individual interviews were conducted with (1) patients who had received the intervention (n=11), (2) secondary care clinicians responsible for delivering the intervention (n=8) and (3) GPs (n=6). Two of the eight secondary care clinicians were actively delivering the intervention (in an in-patient setting), and all GPs provided insights on future implementation. Six patients had received the intervention as a part of inpatient care and five patients provided perspectives on receiving the intervention from outpatient vascular, ulcer or specialised CLTI clinics. Patient participants spoke about living with multiple conditions including diabetes, stroke and amputation and receiving care from primary care (GPs, district nurses, diabetic nurses) secondary care (vascular teams), social care (carers) and family. Three patients were >70 years old but all patients were retired or unemployed and spoke about the impact of PAD on quality of life including managing pain, slow wound healing, poor sleep, reduced daily functioning and limited mobility. Patient characteristics are shown in table 1 and provider characteristics are detailed in table 2.

Of the patient participants contacted by the researcher, (11/26) 42% agreed to take part in an interview (15 people declined) and (8/13) 82% of staff agreed to take part (1 person declined participation and four people did not respond to the invitation). Reasons for non-participation for patients included being unable to remember the intervention (n=5), deteriorating health (n=7), just home from an in-patient admission (n=1), being admitted for angioplasty (n=1) and getting back on with life (n=1). The reason provided by providers for declining participation was moving to a new role and lack of time.

Secondary care clinicians and GPs spoke about all three elements of the intervention and patients provided insights on the education leaflet and GP action letter. The descriptive themes were mapped to implementation theory using the CFIR, to explore connections between the themes, and to compare findings across participant groups.

## Using the CFIR to facilitate analysis

The CFIR comprises five domains that provide an overarching typology of implementation theory (in terms of what works where and why): (1) intervention characteristics (features of the intervention such as complexity), (2) outer setting (the external context that might influence implementation such as health policy and the extent to

**Table 1** Patient participant characteristics

| Patient characteristics n (%) | Patients n=11 |
|---|---|
| **Age** | |
| 41–50 years | 1 (9.1) |
| 51–60 years | 0 (0.00) |
| 61–70 years | 2 (18.2) |
| >70 years | 8 (72.7) |
| **Gender** | |
| Male | 8 (72.7) |
| Female | 3 (27.3) |
| **Ethnicity** | |
| White British | 9 (81.8) |
| White Irish | 1 (9.1) |
| Black/Black British-Caribbean | 1 (9.1) |
| **Marital status** | |
| Single | 2 (18.2) |
| Married | 7 (63.6) |
| Widowed | 2 (18.2) |
| **Time since diagnosis** | |
| >1 year | 3 (27.3) |
| 1–4 years | 4 (36.35) |
| 5–9 years | 0 (0.00) |
| 10 years+ | 4 (36.35) |
| **No of surgical interventions** | |
| None | 1 |
| Angioplasty | 8 |
| Bypass graft | 4 |
| Amputation | 2 |

**Table 2** Provider participant characteristics

| Provider characteristics n (%) | Hospital staff n=8 | GPs n=6 |
|---|---|---|
| **Age** | | |
| 21–30 years | 3 (37.5) | 1 (16.7) |
| 31–40 years | 5 (62.5) | 3 (50.0) |
| 41–50 years | 0 (00.0) | 2 (33.3) |
| **Gender** | | |
| Male | 4 (50.0) | 4 (66.7) |
| Female | 4 (50.0) | 2 (33.3) |
| **Ethnicity** | | |
| White British | 2 (25.0) | 3 (50.0) |
| White other | 1 (12.5) | 1 (16.7) |
| Asian/Asian British-Indian | 2 (25.0) | 2 (33.3) |
| Mixed: white and Asian | 1 (12.5) | 0 (00.0) |
| Chinese | 1 (12.5) | 0 (00.0) |
| Other | 1 (12.5) | 0 (00.0) |
| **Core profession** | | |
| Nurse | 1 (12.5) | 0 (00.0) |
| Vascular surgeon | 7 (87.5) | 0 (00.0) |
| GP trainee | 0 (00.0) | 1 (16.7) |
| GP | 0 (00.0) | 5 (83.3) |
| **Grade** | | |
| Band 8a | 1 (12.5) | 0 (00.0) |
| Senior house officer | 2 (25.0) | 0 (00.0) |
| Specialty trainee | 1 (12.5) | 1 (16.7) |
| Specialty registrar | 2 (25.0) | 0 (00.0) |
| Consultant | 2 (25.0) | 0 (00.0) |
| GP | 0 (00.0) | 5 (83.3) |
| **Time since qualification** | | |
| 0–2 years | 0 (00.0) | 1 (16.7) |
| 2 years + –5 years | 0 (00.0) | 1 (16.7) |
| 5 years + –10 years | 3 (37.5) | 2 (33.3) |
| 10 years + –15 years | 5 (62.5) | 2 (33.3) |
| **Time in current post** | | |
| >6 months | 5 (62.5) | 1 (16.7) |
| 7–12 months | 0 (00.0) | 0 (00.0) |
| 13–24 months | 2 (25.0) | 2 (33.3) |
| 25–36 months | 0 (00.0) | 1 (16.7) |
| 37–48 months | 1 (12.5) | 0 (00.0) |
| 49–60 months | 0 (00.0) | 0 (00.0) |
| 61–75 months | 0 (00.0) | 1 (16.7) |
| 13 years | 0 (00.0) | 1 (16.7) |

GPs, general practitioners.

which patient needs, as well as barriers and facilitators to meet those needs are known and prioritised), (3) inner setting (features of the implementing organisation such as leadership, implementation readiness), (4) characteristics of individuals involved in implementation (such as beliefs and knowledge about the intervention) and (5) implementation process.[13] The themes map to all five overarching CFIR domains across participant groups: intervention characteristics, outer setting, inner setting, characteristics of individuals and process. In summary, for intervention characteristics, all participant groups spoke about the advantage of implementing the care bundle, provided insights on improvements to the design quality and packaging, and acknowledged the need to adapt and tailor the intervention to meet the needs of individuals. Complexity was identified in terms of implementing the intervention across the primary–secondary care interface.

For outer setting, the specific needs of the patient group and barriers and facilitators to meet those needs were highlighted, with all participant groups explaining that a delayed referral to secondary care is a barrier to implementing the intervention. Staff participants went on to speak about external policies and incentives that might influence the spread and uptake of the intervention. For inner setting, the structural characteristics and

the quality of networks and communications within and across each organisation were identified by all groups as influences on implementation. For structural characteristics, participants explained that the ability to follow the advice in the education leaflet is often determined by the availability of supervised exercise programmes, difficulty ordering medication and relying on carers for food shopping. For networks and communications, participants spoke about the need to work as a three-way team, and to have ongoing access to the specialist vascular team. For characteristics of individuals, staff values, attitude, knowledge and skills to provide the intervention and patient readiness to take on the advice of the intervention were highlighted across participant groups. The patient–healthcare provider relationship was identified as a central influence on implementation and there was a particular emphasis on individual stage of change, with staff participants highlighting a need to engage patients in their care and to promote shared decision making. The mapping of themes across CFIR domains and participant groups is included in online supplemental file 2.

A final coding frame for patients, secondary care clinicians and GPs was developed following interpretive analysis. Because of space limitations, a summary of each theme is provided with subthemes being used to define each overarching theme; the full coding framework is included in online supplemental file 3. Four overarching themes were identified: the potential to make a difference, a solution to address the gap in no man's land, prioritising and making it happen and personalised information and supportive conversations for taking on the advice. Similarities and differences between participant groups are presented where they arise.

### Theme 1: the potential to make a difference

GPs and secondary care clinicians felt that the LEGS intervention provides an opportunity to meet patient need and to support implementation of NICE guidance and spoke optimistically about using the intervention to improve patient outcomes and to standardise a minimum quality of care.

> …it standardises the quality of care and standardisation is a big problem that we have in vascular surgery because guidance is so vague. So, I think it raises awareness and standardises that at least we ensure the patient gets a minimum of care (QR15, secondary care clinician, UHL).

GPs and secondary care clinicians went on to explain how the intervention can be used to inform, empower and support relationship building with patients.

> I've gone through the leaflet with them… and knowing that the patient has understood I will spend a couple of minutes going through the leaflet to make sure. If they've got any questions I will try and take the opportunity to build my relationship with the patient

so that they do understand (QR08, secondary care clinician, UHL).

To further improve quality of care, providers identified the intervention as a way to support shared working across primary and secondary care, promoting a team approach and providing an opportunity for collaborative working. One GP explained how the 'patient is in the middle' (QR23) and a secondary care clinician felt that 'we don't sing off the same hymn sheet' (QR08). One GP explained that 'we don't ever see secondary care specialists' and a secondary care clinician explained 'there's always this assumption that the GP will sort it out' (QR13). It was acknowledged that shared working provides an opportunity to understand each other's roles, to manage expectations and to learn from one another. Providers went on to report patient benefits in terms of access to specialist clinicians and working as a multidisciplinary team, 'I've always found that patients who have the opportunity to talk to multiple members of the medical team rather than a doctor or a nurse come out with better outcomes' (QR08, secondary care clinician, UHL).

Patients also spoke about the benefit of joined-up working, with one person stating;

> …you attend lots of different clinics, you've got different people involved in your healthcare. It's not just one thing and everything's linked… it's what they call whole person care. It's not just your blood pressure, it's also the vascular team, it's the podiatrist, it's the diabetes specialist (QR07, out-patient clinic, GSTT).

### Theme 2: a solution to address the gap in no man's land

All participant groups spoke about a need to target patients earlier and identified a service provision gap in preventative medicine and early diagnosis. One secondary care clinician stated;

> I think, almost targeting people earlier on, so claudicants would be really good… claudicants at the very first stage of their peripheral arterial disease, they're not really managed in secondary care, but they're also not really managed in primary care… this gap where they're in no man's land and they almost wait until it's severe enough to managed into secondary care (QR18, secondary care clinician, UHL).

Patients explained how it took time to be referred to secondary care, and in one person's situation, it became an emergency;

> …my doctor prescribed pills and some cream for my foot but they didn't do anything. And then it was only when I went back to podiatry after three months of constant pain, it was a case of just wait and see how things went… I went one day in the November, and they said you need to have your toe off almost immediately… they sent me to [hospital] and [hospital] say 'your toe needs to come off and it needs to come off in 20 minutes time' (QR04, In-patient care, UHL).

Difficulties with patients presenting late, high secondary care referral thresholds resulting in delayed referrals to secondary care and a wait for secondary care to make the diagnosis, alongside lack of support for primary care to commence diagnostics in the community were also listed as barriers to accessing timely care. Some secondary care clinicians felt that the care of those with IC should be managed in primary care.

> I think even the way that if the GP could start the patient, the first medical therapies, like, clopidogrel and statins, if he also could start the patient on exercise therapies. So, it should be implemented at the same kind of wave of the first medical therapy. (QR19, secondary care clinician, UHL).

GPs offered their support and made a request to do more sooner. Some suggested using the intervention in general practice as part of a PAD prevention programme. They acknowledged the value of using the intervention to promote earlier conversations with patients, manage patient expectations and support primary care to start best medical therapy. GPs also spoke about how being involved sooner, and completing the clinical checklist, could be used to support triage of referrals to secondary care. GPs noted that preventative medicine and health promotion is core to their practice, and that sharing information across services and with patients reinforces the message and supports implementation.

### Theme 3: prioritising and making it happen
Secondary care clinicians spoke about the need to prioritise the intervention alongside limited resources and the burden of work. One secondary care clinician explained, 'it shouldn't be seen as an extra thing' (QR08). Individual levels of confidence, attitude, knowledge and experience of the LEGS bundle, or a similar checklist/patient education intervention influenced prioritisation with some staff identifying checklists as a quick and easy-to-use memory prompt. However, another reported that;

> …people feel that it's just a tick-box exercise rather than a safety mechanism, and therefore no engagement with it… [it's] another piece of paper… I think part of the challenge is engagement with clinicians. Clinicians don't like threats to their autonomy… doctors, surgeons in particular are reticent, or certainly some are reticent to engage with checklists because they feel that it is doubling down or they already know this or that, you know, this simplifies things (QR13, secondary care clinician, UHL).

Secondary care clinicians spoke about the extra resources required to deliver the intervention; it takes time to build relationships and to have conversations with patients. One secondary care clinician stated that finding the time to talk would require extra staff, this in an already stretched environment with staff shortages and long waiting times.

> That will take more time, to make it personalised, to make it meaningful… you could almost imagine that they have a consultation and then some of those patients might benefit from sitting down with one of the specialist nurses… you could almost benefit from a 20-minute appointment after the standard clinic appointment to go through all of that. But again, that's a resource burden, that means that you've got to have someone extra in clinic (QR13, secondary care clinician, UHL).

Secondary care clinicians felt that the intervention fits the in-patient environment neatly and could be used as a record of the outpatient clinic consultation. GPs felt that it could support the complexity of assessing patients remotely in the community. The need for training was recognised;

> I think there is a huge lack in primary care education, and I think that goes to show by the starting people coming in are only on like a 40% of them are on best medical therapy so I think a big thing to target would be primary care to inform them (QR18, secondary care clinician, UHL).

GPs went on to request that all primary care staff be involved and not just GPs, and that the primary care training be delivered by the vascular team.

### Theme 4: personalised information and supportive conversations for taking on the advice
Providers and patients spoke about the delivery of the intervention, recognising the importance of offering personalised care and tailoring patient information to individual needs. Secondary care clinicians felt that the intervention should be adapted to identify realistic targets and individual goals, 'or the advice rings hollow' (QR13), and to offer a different approach for each patient. Staff reported;

> I think the hard thing as well is that with the LEGS bundle, sometimes you're giving it to someone who's just had an amputation or about to have an amputation so it's just to be mindful you're talking about things like exercise they look at you very disparagingly and that can be a bit tricky… I think it's a really good checklist and bundle for claudicants and I think that's where it really shines but I think for when you get to the stage of CLTI, I think those individuals often feel that they're too far gone for you to talk to them about lifestyle modification and things like that (QR18, secondary care clinician, UHL).

Patients also spoke about the importance of timing in terms of when they received the lifestyle information.

> If I was a bit younger and I'd got more get-up-and-go, I would gladly have a leaflet and go through it and take up advice and what have you. But as I am at the moment, my health is far from good (QR14, inpatient care, UHL).

…it is a bit too late in that respect so I can only do what I can do. I mean yesterday I was out in the garden doing some weeding but I have to use a seat to do it (QR05, in-patient care, UHL).

Patients explained that they like to receive information face to face and suggested that the education leaflet could be used as a supportive conversation guide. They emphasised the importance of the conversation that goes alongside the advice and information, and the relationship that they have with their health professional(s) in terms of asking questions and being able to talk about making changes like smoking cessation. One patient explained,

… the only thing I've got to do is drink and smoke and watch the tv—cause if I didn't have those three things, I would be pulling my hair out… to be quite honest, I'm quite happy at the moment to be carrying on the way I am. It might sound silly to you, but I enjoy smoking. I love a drink, I like my telly… if you're brow-beaten into packing up when you don't really want to, all this, it can have a great effect on people (QR14, in-patient care, UHL).

GPs and secondary care clinicians acknowledged the need to support patient preference, to involve patients in the conversation and to share decision making.

…you have to strike a balance, and therefore taking the patient's preferences into account is important. We assume that patients want to live the longest life they can, I'm not sure that's always correct. Some patients would rather accept that they live a slightly shorter life but with the benefit of enjoying certain things that we all know are less healthy (QR13, secondary care clinician, UHL).

It was acknowledged that everyone knows they need to stop smoking and it is easy to bombard people with information. Solutions such as motivational interviewing techniques were identified by one GP as a way to support engagement conversations. Secondary care clinicians felt that providing information on the benefits of following the advice vs taking pleasure away might help. For others, frank conversations that explain the risks help;

I think honesty is the most important aspect of this and bringing people in front of the data, highlighting the data, and actually not sugar coating anything. We know, for example, that if you have critical ischemia, so severe peripheral arterial disease, if your prognosis is worse, your five year survival is worse than some kinds of cancer. So, if you tell someone that unless you actually take this intervention seriously, your likelihood of not being around in five years is very high. I know it sounds bleak, but it may be the wake-up call that someone needs. Or explaining that from the moment you have severe arterial peripheral disease, the likelihood of you losing your leg and being wheelchair bound may be an important thing to highlight as opposed to speaking in a circumspect way. I think

just being honest and explaining the risks may be a way of engaging patients. (QR15, secondary care clinician, UHL).

A patient explained how the honest frank information helped;

The doctor actually threatened me. He said to me, if I'm going to do this operation, you got to stop smoking. So, I did. That helped, or I could lose my leg. I thought it was brilliant the way [the doctor] come out with it and I've never stopped thanking him for it. It works (QR06, out-patient care, GSTT).

## DISCUSSION

The aim of the study was to provide an understanding of the feasibility and implementation of the LEGS (LEaflet Gp letter Structured checklist) intervention, a complex intervention in the form of a healthcare bundle, designed to improve the care of patients with PAD in the NHS. Our inductive approach to analysis found four overarching themes: (1) the potential to make a difference, (2) a solution to address the gap in no man's land, (3) prioritising and making it happen and (4) personalised information and supportive conversations for taking on the advice. These themes provide an overarching message of acceptability for the LEGS intervention by highlighting benefits at the individual and system level and by outlining strategies for increasing impact and uptake. We used the Consolidated Framework of Implementation Research (CFIR) to guide our analysis, and all five CFIR domains are represented.[13] The CFIR is well placed for developing a model for implementing secondary care-initiated treatment in primary care.[17]

All participants spoke positively about using the intervention to make a difference to patient care, as an opportunity to meet patient needs, and as a way to enhance the quality of PAD care through improved working practices. This promotes a streamlined patient pathway from primary care to secondary care and to shared care. This reflects the value of an agreed pathway of care and shared partnership working that provides an opportunity for the intervention to be delivered as part of a wider team and to prioritise patient-centred care.[17]

There was a consensus of optimism in participant accounts about the potential to make a difference. While barriers to seeking care for those with PAD are well-documented,[18 19] the findings confirm that the period between diagnosis of early-stage PAD (eg, claudication) and subsequent potential diagnosis of advanced PAD with severe symptoms, is a time of no man's land, when patients do not receive any help, education, support or appropriate medication to address their condition.[9] As such, the benefits of the LEGS intervention were identified in terms of meeting the needs of this subgroup of patients, and to address the gap in care. GPs were optimistic that they can deliver the LEGS intervention

with training and ongoing sustained communication with secondary care. The findings highlight the need to consider training requirements and systems for maintaining open primary–secondary care communication to support implementation. For example, education on tailoring the intervention for patients at both early and late stages of PAD that can be used across primary and secondary care settings, similar to the model used with other chronic conditions[17 20] Improving the awareness of PAD among healthcare providers and supporting patients to take responsibility for their own health are therefore important determinants in the success of implementation.[21]

While the three elements of the LEGS intervention have been designed to facilitate interaction between secondary and primary care following diagnosis of PAD, the intervention focus until now, has been on secondary care services, that is, hospital admission or outpatient clinic. However, our findings indicate that the intervention can be used to address the gap in service provision for patients who do not yet meet the referral criteria for secondary care services. We; therefore, propose to extend the use of the LEGS intervention across the primary-secondary care interface. Our findings also confirm the need for individualised patient care, supportive communication and shared decision making in order to take on the advice in a more effective way.[22] The finalised LEGS intervention should therefore promote supportive conversation between patient and healthcare staff, both in primary and secondary care settings. The open approach to changing lifestyle and behaviour could be built on techniques like motivational interviewing, which is already commonly used in primary care practice.[23] Lastly, our findings confirm that healthcare providers are optimistic and positive about the introduction of an intervention which will help them to deliver more consistent and guideline-based PAD treatment. The challenge now is to take this enthusiasm together with the lessons learnt from this qualitative research to adapt the LEGS intervention before wider testing and adoption in the NHS. The main changes will relate to usability in primary care after diagnosis of early-stage PAD, facilitation of effective communication not only between primary and secondary care staff, but also patients and staff, as well as focusing on specific needs of patients by prioritising areas to focus on in an incremental or stepwise manner.

## Strengths and limitations

The paper adds new knowledge on the development and implementation of a community and hospital intervention for PAD. It is important to note that the findings are specific to the CHABLIS study. It is possible, however, to enhance transferability by describing the research context and assumptions, and by making connections between the analysis of participants accounts and claims in the extant literature.

While the sample size for this study may seem small, the volume of interview data we collected is unique for this patient population. Many factors influence the success of recruiting participants in research and engaging vulnerable patient populations brings additional complexity.[24] PAD is a chronic and progressive disease with burdensome symptoms including physical disability, cardiovascular risks and mortality.[25] Those who live with PAD are vulnerable because they are living with extreme frailty, often with the threat or result of limb loss, and with risks associated with poorer socioeconomic conditions.[26] Alongside, patients with PAD might be less receptive to engaging with research because they experience specific barriers to accessing healthcare meaning that taking part in research alongside managing healthcare and health needs might be considered onerous.[19]

A strength is the focus on rigour and triangulation of participant perspectives. We also acknowledged the complexity of intervention implementation across the secondary-primary care interface and therefore used a systems research perspective for addressing implementation at the outset of the research. We used implementation theory to guide analysis and to provide a layered coding approach, to ensure that intervention and implementation recommendations are novel, grounded in the data and useful.[13]

While the anonymity of telephone use can allow participants to disclose sensitive information, telephone interviews have received criticism for compromising interviewer/participant rapport and interaction, and for limiting contextual data due to the absence of face-to-face contact and visual cues. However, this method of data collection is convenient, in that it is flexible (in terms of time and location), accessible (ie, remote research conducted during the COVID-19 pandemic), and allows for a wide reach (eg, accessing a diverse population from both urban and rural areas).[27]

## Conclusions

The LEGS intervention can be tailored for use at early and late stages of PAD, can be provided across primary and secondary care settings, and provides an opportunity to promote shared working across the primary–secondary care interface. These findings are important to the specifics of the CHABLIS study but can also go some way in informing the wider learning for other interventions that are implemented across the primary–secondary care interface.

**Author affiliations**
[1]Department of Health Services & Population Research, King's College London, London, UK
[2]Division of Methodologies, Florence Nightingale Faculty of Nursing, Midwifery & Palliative Care, King's College London, London, UK
[3]University of Leicester, Leicester, UK
[4]Guy's and St Thomas' NHS Foundation Trust, London, UK
[5]University of Birmingham, Birmingham, UK
[6]NIHR Cardiovascular Sciences, University of Leicester, Leicester, UK

**Acknowledgements** We would like to thank all patients, secondary care staff and GPs who participated in this study and who generously gave their time and honest thoughts. We would like to thank the local research teams for their

support in recruiting to this qualitative study and we are also grateful to the NIHR Clinical Research Network Research Delivery Managers that helped to recruit GP participants. We would also like to thank the stakeholder PPI group for their time and input to the CHABLIS study.

**Contributors** CL contributed to the qualitative study design, coordinated the study, led data collection and analysis, and drafted the manuscript. AS is principal investigator to the CHABLIS study, acquired funding, participated in the study design, contributed to data collection, and drafted the manuscript. PS contributed to data collection and reviewed the manuscript. RB and BB drafted the manuscript. EW contributed to data collection and reviewed the manuscript. VL is guarantor, designed the qualitative study, provided oversight on study coordination, data collection and analysis, and drafted the manuscript. All authors read and approved the final manuscript.

**Funding** This article presents independent research funded by National Institute for Health Research for Patient Benefit (RfPB) Programme (funder's reference: NIHR202008).

**Disclaimer** The views expressed are those of the author(s) and not necessarily those of the NIHR or the Department of Health and Social Care.

**Competing interests** None declared.

**Patient and public involvement** Patients and/or the public were involved in the design, or conduct, or reporting, or dissemination plans of this research. Refer to the Methods section for further details.

**Patient consent for publication** Consent obtained directly from patient(s)

**Ethics approval** The research was reviewed and approved by the NHS Wales Research Ethics Committee (REC), reference number 20/WA/0319 and the NHS Health Research Authority (HRA).

**Provenance and peer review** Not commissioned; externally peer reviewed.

**Data availability statement** No data are available. This study uses data (containing potentially identifying and/or sensitive information) collected from a small group of staff participants and a vulnerable patient population, and involves indirect identifiers (such as sex, ethnicity, location) that may risk the identification of study participants. Sharing data outside of the anonymised excerpts and quotations included in the paper will violate the agreement to which the participants consented.

**ORCID iDs**
Clair Le Boutillier http://orcid.org/0000-0003-3665-0315
Vanessa Lawrence http://orcid.org/0000-0001-7852-2018

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
