## [Reviewer comments · BMJ Open]

ARTICLE DETAILS

TITLE (PROVISIONAL)	Factors that influence the feasibility and implementation of a complex intervention to improve the treatment of Peripheral Arterial Disease in primary and secondary care: a qualitative exploration of patient and provider perspectives
AUTHORS	Le Boutillier, Clair; Saratzis, Athanasios; Saha, Prakash; Benson, Ruth; Bridgwood, Bernadeta; Watson, Emma; Lawrence, Vanessa

VERSION 1 – REVIEW

REVIEWER	Ouyang, Menglu The George Institute for Global Health
REVIEW RETURNED	15-Aug-2022

GENERAL COMMENTS	This qualitative research explored the acceptability and implementation of a bundle to improve PAD at both primary and secondary level of care. Both patients and implementers (clinicians and GPs) were interviewed, which provided a comprehensive perspectives from the stakeholders. My comments: 1. Are there any framework or theory to guide this study? The MRC guidance is helpful to understand the implementation of a complex intervention. There are several indicators should be explored to understand the implementation not only acceptability but fidelity, sustainability and adoption.2. Some contextual factors such as health system and policy might also influence the implementation. It will be good to have a theme on these factors.3. I recommend to revise the Results based on the objectives by using subheadings such as acceptability, barriers and facilitators.4. Since the theme titles are ambiguous, the subthemes will be more helpful to understand the results more in details, which are important to address the aims of this study. I recommend to move the characteristics of patients and clinicians to supplementary to save space.
---

REVIEWER	Chudyk, Anna University of Manitoba
REVIEW RETURNED	19-Aug-2022

GENERAL COMMENTS	Thank you for the opportunity to review this article on patient-provider factors that influence the implementation of a community and hospital care bundle. Strengths of the article include its comprehensive approach to analysis and the gathering of different stakeholder perspectives to meet the research aim. However, I have highlighted some concerns that need to be addressed prior to the article being suitable for publication. As you will see, they primarily pertain to lack of congruence among the research aims
---

	stated throughout the study, inadequate detail provided around recruitment and participant flow, lack of clarity at the beginning of the results section, and the need to expand the discussion a little to better describe how the study fits into the context of the broader literature. Lines 120-122: the information you provide in the second half of the sentence regarding LEGS is redundant to the information that you provided in the entire previous paragraph. Please revise perhaps to "...feasibility and effectiveness of the LEGS intervention." Lines 124-128: your stated objectives don't clearly match up with the ones you provide in the abstract (acceptability and feasibility). Lines 132-133: the stated desired insights don't clearly (evidently, intuitively) match up with the objectives listed in lines 124-128 and the abstract. Consider starting more broad and then narrowing in on specific objectives to improve flow and congruence. I think that what would help once you decide upon the languaging of your aims is to define for the reader what you mean by, for example, feasibility or acceptability (or whatever your aims are). Methods: Please state that you use the COREQ checklist to guide your reporting. Methods: Please state whether participants were compensated for taking part in the study Lines 144-148: can you please clarify these stakeholders' role in the intervention? It is clear that the first two groups were directly involved with the intervention (patient/carer) but I'm unclear on the relationship between these stakeholders and the intervention that was delivered... Lines 153: Is this who happened to be recruited or were these your recruitment criteria? Be clearer. Lines 154-161: provide the number of patients who received the intervention (and thus made up your sampling frame). Same goes re: providing numbers for the number of 2ndary care providers and GPs eligible to participate. I think it makes sense to put these numbers here because you state "all eligible to participate were approached..." Lines 157-161: how exactly were the secondary care clinicians invited (e.g., email or...)? You need more details that would help others reproduce your study. Lines 150-168: it is unclear how you chose who you approached and when/how you decided to stop recruitment for each participant group... did you approach every eligible participant? Randomly select participants until you reached the desired number of participants, etc. Lines 172-174: I think it makes more sense to refer to supplement 1 (interview guide) in the preceding sentence as I initially thought that supplement 1 contained the interview schedule (based on where you refer to it). Data collection: Provide some high-level detail about the types of questions that were asked to save readers from having to go through the interview guides unless they want more in depth information. Lines 187: it is unclear what the (CL) refers to Line 188-190: It would be nice if you expanded upon the CFIR framework in 1-2 sentences to provide further context for readers unfamiliar with it. Data analysis – provide further detail (even just brackets with the initials) re: who carried out the steps. Line 197-198: expand upon what you mean by "in a way that supported data analysis" or consider stopping the sentence after "... manage the data." Lines 198-200: provide more detail about the coding that was done by the second rater as their role is unclear. Lines 202-205: consider using the IAP2 engagement spectrum or its adapted version (https://journals.plos.org/plosone/article?id=10.1371/journal.pone.0193579) to better describe PPI in developing the intervention and contributions to
--	--

	interpretation of study findings. Consider also providing a bit more information about the development sessions and also how the PPI co-applicant actually helped interpret the findings (via face-to-face meetings, providing written feedback on the themes, etc.). Lines 229-242: This information should be contained in the first paragraph of the results and also summarized in a diagram that details the flow of participants into the study □ e.g., the number of eligible participants by group, number approached by group, number refused to participate (and reasons for refusal (if you have them) by group, and number agreeing to participate by group. Line 229: remove (CL) as the information about who did the contacting belongs in the methods Lines 229-231: report n/n (%) related to who agreed to participate... n/n (42%), n/n (82%). Lines 235: The sentence “secondry care clinicians and GPS spoke...” should be a new paragraph as you are no longer describing the flow of participants into the study and should appear after the paragraph describing the participants. Give this paragraph a subheading to cue the reader that you’ve moved from describing the flow of participants into the study and participant characteristics to presentation of the the themes. Lines 235-242: I find this information confusing. Likely partially because of it being tucked in with the “flow of participants” info and maybe due to the lack of explanation about the CFIR in the methods. I don’t have a concrete suggestion around how to clarify what’s here but it does need revision. Perhaps including (and not limited to) some added detail in the methods that better sets the reader’s expectations and understanding around the information presented here. You definitely need to provide some explanation of what the CFIR domains mean as this understanding is foundational to understanding what you present in the rest of your results. Lines 244-262: It would be nice if you could create a figure that summarizes this info and is embedded in the manuscript (instead of a supplement). It will help the reader better absorb findings you present here. Lines 272 onwards: I really like the way you present your findings across the themes (e.g., use of headings and subheadings, italics, placement of quotes). Lines 450: Ensure the aim you report here is consistent with your revision based on my initial comments re: fluctuating aims. Lines 460-461: Your use of moreover makes this read as an incomplete sentence. Consider merging this paragraph’s first two sentences since I think your intention is to list what all participants spoke positively about. Discussion: I really like your focus on the lessons learned and next steps in your discussion, but I think you are missing a paragraph that explores how your findings fit in with the broader literature. Line 506: PAD not Peripheral Arterial Disease Strengths and Limitations: I usually place these before my conclusions. I'm unsure if BMJ Open has a preference for this ordering.
--	--

VERSION 1 – AUTHOR RESPONSE

- Are there any frameworks or theory to guide this study? The MRC guidance is helpful to understand the implementation of a complex intervention. There are several indicators should be explored to understand the implementation not only acceptability but fidelity, sustainability and adoption.

Thank you for this important point. We used the MRC framework to guide the research question and objectives of the study and reference Skivington et al., 2021. We have now made this explicit in our writing to add clarity (Lines 124-132):

The aim of the qualitative study was to gain an understanding of the feasibility and implementation of the LEGS intervention, by exploring patient and provider perspectives on receiving and delivering the intervention (Skivington et al., 2021). Guided by the MRC framework for developing and evaluating complex interventions, the objectives of the qualitative study were to 1) to explore patient perspectives on receiving the LEGS intervention, 2) to gather secondary care clinician views on feasibility, and to understand factors that influence the success of implementation, and 3) to identify barriers and facilitators that influence the success of future implementation from the GP perspective.

3. Some contextual factors such as health system and policy might also influence the implementation. It will be good to have a theme on these factors.

We report (Lines 208-211) that we used existing implementation theory (the Consolidated Framework for Implementation Research (CFIR)) to facilitate analysis and to provide a layered coding approach. The CFIR provides an overarching typology (39 constructs) of implementation theory in terms of works where and why across multiple contexts, and this includes health system and policy influences (outer setting). We have added additional detail on the CFIR and explain that factors such as health system and policy might have an influence on implementation (called the outer setting) (Lines 264-271):

The CFIR comprises five domains that provide an overarching typology of implementation theory (in terms of what works where and why): i) intervention characteristics (features of the intervention such as complexity), ii) outer setting (the external context that might influence implementation such as health policy and the extent to which patient needs, as well as barriers and facilitators to meet those needs are known and prioritised), iii) inner setting (features of the implementing organisation such as leadership, implementation readiness), iv) characteristics of individuals involved in implementation (such as beliefs and knowledge about the intervention), iv) implementation process...

For outer setting, the specific needs of the patient group needs and barriers and facilitators to meet those needs were highlighted, with all participant groups explaining that a delayed referral to secondary care is a barrier to implementing the intervention. Staff participants went on to speak about external policies and incentives that might influence the spread and uptake of the intervention. The mapping of themes across CFIR domains and participant groups is also included in Online Data Supplement 2.

4. I recommend to revise the Results based on the objectives by using subheadings such as acceptability, barriers and facilitators.

We approached the data inductively and used the CFIR to place our findings in the context of implementation theory. The CFIR is intended to be flexible in application so that researchers can tailor the framework to the specific intervention and context being studied. We used the CFIR headings to map our descriptive themes and have remained close to the data with theme derivation/names rather than imposing subheadings. This comprehensive approach to qualitative analysis has been commended as a strength of the study by other reviewers. We hope this clarifies our rationale for the choice of subheadings.

5. Since the theme titles are ambiguous, the subthemes will be more helpful to understand the results more in details, which are important to address the aims of this study. I recommend to move the characteristics of patients and clinicians to supplementary to save space.

We have revised the text to explain that the content of the sub-themes is included in the paper by way of being used to summarise the overarching theme. They are not however labelled with separate headings in the text (Lines 307-309):

Because of space limitations, a summary of each theme is provided with subthemes being used to define each overarching theme.

6. Lines 120-122: the information you provide in the second half of the sentence regarding LEGS is redundant to the information that you provided in the entire previous paragraph. Please revise perhaps to "...feasibility and effectiveness of the LEGS intervention."

Thank you for bringing this to our attention. We have made this change (Lines 122-124).

CHABLIS is a multicentre prospective mixed-methods cohort study that aims to determine the feasibility and effectiveness of the LEGS intervention.

7. Lines 124-128: your stated objectives don't clearly match up with the ones you provide in the abstract (acceptability and feasibility).

Thank you for bringing this to our attention. We have amended the text (Lines 37-40):

Our aim was to examine the feasibility and implementation of a complex intervention to improve the care of patients with Peripheral Arterial Disease (the LEGS intervention) from the perspective of patients, general practitioners and secondary care clinicians.

8. Lines 132-133: the stated desired insights don't clearly (evidently, intuitively) match up with the objectives listed in lines 124-128 and the abstract. Consider starting more broad and then narrowing in on specific objectives to improve flow and congruence. I think that what would help once you decide upon the languaging of your aims is to define for the reader what you mean by, for example, feasibility or acceptability (or whatever your aims are).

We have amended the text which now reads (Lines 136-138):

The study used qualitative methods to provide insights into the feasibility and acceptability of the intervention as well as the complexities of implementation (Skivington et al., 2021).

In terms of languaging, we used the MRC framework to guide the research question and objectives of the study. We have now made this explicit in our writing to add clarity (Lines 124-132):

The aim of the qualitative study was to gain an understanding of the feasibility and implementation of the LEGS intervention, by exploring patient and provider perspectives on receiving and delivering the intervention (Skivington et al., 2021). Guided by the MRC framework for developing and evaluating complex interventions, the objectives of the qualitative study were to 1) to explore patient perspectives on receiving the LEGS intervention, 2) to gather secondary care clinician views on feasibility, and to understand factors that influence the success of implementation, and 3) to identify barriers and facilitators that influence the success of future implementation from the GP perspective.

9. Methods: Please state that you use the COREQ checklist to guide your reporting.

We have included this information in Lines 142-144:

The consolidated criteria for reporting qualitative research (COREQ) guideline was used to ensure quality when reporting the research (Tong, Sainsbury, & Craig, 2007).

10. Methods: Please state whether participants were compensated for taking part in the study

We have now provided this detail in Line 182-183:

Participants were not reimbursed for participating in the research.

11. Lines 144-148: can you please clarify these stakeholders' role in the intervention? It is clear that the first two groups were directly involved with the intervention (patient/carer) but I'm unclear on the relationship between these stakeholders and the intervention that was delivered...

We have provided additional information to aid clarity (Lines 148-151 and 153-157):

... semi-structured individual interviews were conducted with secondary care clinicians who were responsible for delivering (that is, delivering or intending to deliver) the intervention, including junior doctors, vascular nurses, and vascular surgeons.

... semi-structured individual interviews were conducted with GPs to gain an understanding of the barriers and facilitators to future implementation in primary care and across the NHS. GPs were not involved in delivering the study intervention and were asked to report on perceived factors that might influence implementation beyond the context of the study.

We also report in the results section (Lines 234-236) that:

Two of the eight secondary care clinicians were actively delivering the intervention (in an in-patient setting), and all GPs provided insights on future implementation.

12. Lines 154-161: provide the number of patients who received the intervention (and thus made up your sampling frame). Same goes re: providing numbers for the number of 2ndary care providers and GPs eligible to participate. I think it makes sense to put these numbers here because you state "all eligible to participate were approached..."

We have added details and provide the numbers as requested (Lines 169-180):

All patients who had received the intervention and who gave consent for the researcher to make contact about the qualitative study were approached for participation (n=26). All secondary care clinicians responsible for delivering the intervention (n=13) were invited to take part. Secondary care clinicians were invited to participate following an in-person meeting with the local principal investigator at each site. A participant information sheet that outlined the study was provided to all potential participants with time to consider whether or not they would like to take part. GP participants were recruited from Clinical Commissioning Groups linked to the two secondary care implementation sites, chosen to enhance joined up working and to provide a local perspective on implementation across community and hospital settings. Potential GP participants (n=7) were first approached by the local Clinical Research Network research delivery manager or local contacts, and subsequently recruited by the lead author via telephone or email.

13. Lines 157-161: how exactly were the secondary care clinicians invited (e.g., email or...)?

We have added more detail (Lines 172-175):

Secondary care clinicians were invited to participate following an in-person meeting with the local principal investigator at each site. A participant information sheet that outlined the study was provided to all potential participants with time to consider whether or not they would like to take part.

14. Lines 150-168: it is unclear how you chose who you approached and when/how you decided to stop recruitment for each participant group... did you approach every eligible participant? Randomly select participants until you reached the desired number of participants, etc.

We have added text to clarify (Lines 169-172):

All patients who had received the intervention and who gave consent for the researcher to make contact about the qualitative study were approached for participation (n=26). All secondary care clinicians responsible for delivering the intervention (n=13) were invited to take part.

15. Lines 172-174: I think it makes more sense to refer to supplement 1 (interview guide) in the preceding sentence as I initially thought that supplement 1 contained the interview schedule (based on where you refer to it).

We have made this change (Lines 189-194):

Interviews used flexible open-ended questions for early data collection to gather a rich and detailed understanding of participants' perspectives (included in Online Data Supplement 1). The patient and provider interview schedules were revised iteratively in response to the priorities and concerns of participants and gathered information on factors that facilitate or hinder implementation and on possible solutions to overcome any identified barriers.

16. Data collection: Provide some high-level detail about the types of questions that were asked to save readers from having to go through the interview guides unless they want more in depth information.

We have added this detail (Line 191-194):

The patient and provider interview schedules were revised iteratively in response to the priorities and concerns of participants and gathered information on factors that facilitate or hinder implementation and on possible solutions to overcome any identified barriers.

17. Lines 187: it is unclear what the (CL) refers to

Thank you for bringing this to our attention. We have deleted the (CL).

18. Line 188-190: It would be nice if you expanded upon the CFIR framework in 1-2 sentences to provide further context for readers unfamiliar with it.

Many thanks for this. We have added information to describe the CFIR (Lines 264-271):

The CFIR comprises five domains that provide an overarching typology of implementation theory (in terms of what works where and why): i) intervention characteristics (features of the intervention such as complexity), ii) outer setting (the external context that might influence implementation such as health policy and the extent to which patient needs, as well as barriers and facilitators to meet those needs are known and prioritised), iii) inner setting (features of the implementing organisation such as leadership, implementation readiness), iv) characteristics of individuals involved in implementation (such as beliefs and knowledge about the intervention), iv) implementation process

19. Data analysis – provide further detail (even just brackets with the initials) re: who carried out the steps.

We have clarified (Lines 218-220):

The lead author (CL) directed and conducted the analysis. Coding by a second rater (VL) was undertaken to provide an opportunity to reflect on the coding approach, and to enhance the interpretive depth of the data

20. Line 197-198: expand upon what you mean by “in a way that supported data analysis” or consider stopping the sentence after “... manage the data.”

We have amended the text and stopped the sentence after "...manage the data" (Line 216-217):

Data collection occurred concurrently with data analysis; NVivo QSR International qualitative analysis software (version 12) was used to manage the data.

21. Lines 198-200: provide more detail about the coding that was done by the second rater as their role is unclear.

We have added more information to provide clarity (Lines 218-222):

Coding by a second analyst (VL) was undertaken to provide an opportunity to reflect on the coding approach, and to enhance the interpretive depth of the data (13). This involved an independent review of the data, the themes, and the CFIR mapping/tabulation. The overarching coding framework was developed following discussion with both analysts.

22. Lines 202-205: consider using the IAP2 engagement spectrum or its adapted version to better describe PPI in developing the intervention and contributions to interpretation of study findings. Consider also providing a bit more information about the development sessions and also how the PPI co-applicant actually helped interpret the findings (via face-to-face meetings, providing written feedback on the themes, etc.).

Thank you for bringing this to our attention to this. We have used the research-relevant modified IAP2 spectrum to further describe the PPI contributions (Lines 225-229):

The LEGS intervention and CHABLIS mixed-methods study design were developed in collaboration with a stakeholder PPI group that included patients, family members and carers, and clinicians working in primary and secondary care (Bammer, 2019).

We have removed the information about the development sessions and analysis as these are specific to developing the intervention – and not to the qualitative study – and will be reported elsewhere (paper in preparation).

23. Lines 229-242: This information should be contained in the first paragraph of the results and also summarized in a diagram that details the flow of participants into the study e.g., the number of eligible participants by group, number approached by group, number refused to participate (and reasons for refusal (if you have them) by group, and number agreeing to participate by group.

We have provided the number of potential participants in the methods for clarity (Lines 169-179). We have reported the number of people who declined to participate (and reasons for refusal, and the number agreeing to participate by group in the results section (Lines 250-256). We have not included a flow diagram of recruitment because our intention is to focus the paper on the experiences (rather than the numbers) of the included participants. We have discussed the limitations involved in recruiting patients with PAD in research in the discussion (Lines 562-572) which reads:

While the sample size for this study may seem small, the volume of interview data we collected is unique for this patient population. Many factors influence the success of recruiting participants in research and engaging vulnerable patient populations brings additional complexity. PAD is a chronic and progressive disease with burdensome symptoms including physical disability, cardiovascular risks and mortality. Those who live with PAD are vulnerable because they are living with extreme frailty, often with the threat or result of limb loss, and with risks associated with poorer socio-economic conditions. Alongside, patients with PAD might be less receptive to engaging

with research because they experience specific barriers to accessing healthcare meaning that taking part in research alongside managing health care and health needs might be considered onerous.

24. Line 229: remove (CL) as the information about who did the contacting belongs in the methods

We have deleted the (CL).

25. Lines 229-231: report n/n (%) related to who agreed to participate... n/n (42%), n/n (82%).

We have added this detail (Lines 250-252):

Of the patient participants contacted by the researcher, (11/26) 42% agreed to take part in an interview (fifteen people declined) and (8/13) 82% of staff agreed to take part (1 person declined participation and 4 people did not respond to the invitation).

26. Lines 235: The sentence “secondary care clinicians and GPS spoke...” should be a new paragraph as you are no longer describing the flow of participants into the study and should appear after the paragraph describing the participants. Give this paragraph a subheading to cue the reader that you’ve moved from describing the flow of participants into the study and participant characteristics to presentation of the themes.

We have moved the text to a new paragraph and provided a sub-heading: Using the CFIR to facilitate analysis (Line 263).

27. Lines 235-242: I find this information confusing. Likely partially because of it being tucked in with the “flow of participants” info and maybe due to the lack of explanation about the CFIR in the methods. I don’t have a concrete suggestion around how to clarify what’s here but it does need revision. Perhaps including (and not limited to) some added detail in the methods that better sets the reader’s expectations and understanding around the information presented here. You definitely need to provide some explanation of what the CFIR domains mean as this understanding is foundational to understanding what you present in the rest of your results.

We have added more detail on the CFIR, the CFIR domains and our rationale for using the framework (Lines 263-304):

Using the CFIR to facilitate analysis

The CFIR comprises five domains that provide an overarching typology of implementation theory (in terms of what works where and why): i) intervention characteristics (features of the intervention such as complexity), ii) outer setting (the external context that might influence implementation such as health policy and the extent to which patient needs, as well as barriers and facilitators to meet those needs are known and prioritised), iii) inner setting (features of the implementing organisation such as leadership, implementation readiness), iv) characteristics of individuals involved in implementation (such as beliefs and knowledge about the intervention), iv) implementation process (Damschroder et al., 2009). The themes map to four of the five overarching CFIR domains across participant groups: intervention characteristics, outer setting, inner setting, and characteristics of individuals. In summary, for intervention characteristics, all participant groups spoke about the advantage of implementing the care bundle, provided insights on improvements to the design quality and packaging, and acknowledged the need to adapt and tailor the intervention to meet the needs of individuals. Complexity was identified in terms of implementing the intervention across the primary-secondary care interface. For outer setting, the specific needs of the patient group needs and barriers and facilitators to meet those needs were highlighted, with all participant groups explaining that a delayed referral to secondary care is a barrier to implementing the intervention. Staff participants went on to

speaking about external policies and incentives that might influence the spread and uptake of the intervention. For inner setting, the structural characteristics and the quality of networks and communications within and across each organisation were identified by all groups as influences on implementation. For structural characteristics, participants explained that the ability to follow the advice in the education leaflet is often determined by the availability of supervised exercise programmes, difficulty ordering medication, and relying on carers for food shopping. For networks and communications, participants spoke about the need to work as a three-way team, and to have ongoing access to the specialist vascular team. For characteristics of individuals, staff values, attitude, knowledge, and skills to provide the intervention and patient readiness to take on the advice of the intervention were highlighted across participant groups. The patient-healthcare provider relationship was identified as a central influence on implementation and there was a particular emphasis on individual stage of change, with staff participants highlighting a need to engage patients in their care and to promote shared decision making. The mapping of themes across CFIR domains and participant groups is included in Online Data Supplement 2.

28. Lines 244-262: It would be nice if you could create a figure that summarizes this info and is embedded in the manuscript (instead of a supplement). It will help the reader better absorb findings you present here.

We attempted to create a figure to illustrate the overarching themes but there is complexity in that the CFIR was used as a part of the process - to provide a layered approach to coding (by mapping descriptive themes to implementation theory). The overarching themes are the final result of that process and an additional level of interpretation. The interpretive analysis allowed for the refining of the specifics of themes and thematic patterns generated by the data, and so the CFIR domains are integrated within them and do not map directly to any one theme.

We describe the coding process in Lines 203-214:

Data analysis began with repeated re-reading of individual transcripts and re-listening of sound files for data immersion. This was followed by line-by-line open coding, where extracts were coded under one or several descriptive themes, that is organising the data according to semantic content, to capture their meaning and reflect the content of the data. Each theme was refined, and where data allowed, further sub-themes were developed. The Consolidated Framework for Implementation Research (CFIR) was then used as a matrix to organise the early themes, to explore patterns and relationships in the data, and to map connections between the themes and the three participant groups. Lastly, interpretive analysis allowed for the refining of the specifics of themes and thematic patterns, and a final coding frame for patients and providers was developed. Refinements to the specifics of themes, and thematic patterns continued until a useful and meaningful analysis was achieved.

29. Lines 450: Ensure the aim you report here is consistent with your revision based on my initial comments re: fluctuating aims.

Many thanks. We have provided clarity and ensured the aim is consistent with our earlier revision (Lines 494-497):

The aim of the study was to provide an understanding of the feasibility and implementation of the LEGS (LEaflet Gp letter Structured checklist) intervention, a complex intervention in the form of a healthcare bundle, designed to improve the care of patients with PAD in the NHS.

30. Lines 460-461: Your use of moreover makes this read as an incomplete sentence. Consider merging this paragraph's first two sentences since I think your intention is to list what all participants spoke positively about.

Many thanks for highlighting this. We have amended the text to read (Lines 504-506):

All participants spoke positively about using the intervention to make a difference to patient care, as an opportunity to meet patient needs, and as a way to enhance the quality of PAD care through improved working practices.

31. Discussion: I really like your focus on the lessons learned and next steps in your discussion, but I think you are missing a paragraph that explores how your findings fit in with the broader literature.

We have added further detail on how the findings fit with the broader literature to the discussion (Lines 500-526):

We used the Consolidated Framework of Implementation Research (CFIR) to guide our analysis, and all five CFIR domains are represented (Damschroder et al., 2009). The CFIR is well-placed for developing a model for implementing secondary care-initiated treatment in primary care (Le Boutillier et al., 2022). All participants spoke positively about using the intervention to make a difference to patient care, as an opportunity to meet patient needs, and as a way to enhance the quality of PAD care through improved working practices. This promotes a streamlined patient pathway from primary care to secondary care and to shared care. This reflects the value of an agreed pathway of care and shared partnership working that provides an opportunity for the intervention to be delivered as part of a wider team and to prioritise patient-centred care (Le Boutillier et al., 2022). There was a consensus of optimism in participant accounts about the potential to make a difference. While barriers to seeking care for those with PAD are well-documented (Tan et al., 2022; Meffen et al., 2021), the findings confirm that the period between diagnosis of early-stage PAD (e.g., claudication) and subsequent potential diagnosis of advanced PAD with severe symptoms, is a time of no man's land, when patients do not receive any help, education, support, or appropriate medication to address their condition (Saratzis et al., 2019). As such, the benefits of the LEGS intervention were identified in terms of meeting the needs of this subgroup of patients, and to address the gap in care. GPs were optimistic that they can deliver the LEGS intervention with training and ongoing sustained communication with secondary care. The findings highlight the need to consider training requirements and systems for maintaining open primary-secondary care communication to support implementation. For example, education on tailoring the intervention for patients at both early and late stages of PAD, that can be used across primary and secondary care settings, similar to the model used with other chronic conditions (Smith et al., 2017; Le Boutillier et al., 2022). Improving the awareness of PAD among healthcare providers and supporting patients to take responsibility for their own health are therefore important determinants in the success of implementation (Bridgwood et al., 2020).

32. Line 506: PAD not Peripheral Arterial Disease

We have made this change (now Line 558)

33. Strengths and Limitations: I usually place these before my conclusions. I'm unsure if BMJ Open has a preference for this ordering.

Many thanks. We have made this change.

We have also updated our author affiliations. We hope that these responses and improvements make the paper suitable for publication in BMJ Open.

Yours faithfully,

Clair Le Boutillier

VERSION 2 – REVIEW

REVIEWER	Ouyang, Menglu The George Institute for Global Health
REVIEW RETURNED	01-Nov-2022

GENERAL COMMENTS	Thanks for the responses and revisions for this piece of work. I have some minor comments: 1. In the first paragraph of discussion, is it possible to speak out the main points from the findings, instead of generally describe the themes titles? It will be more straightforward and helpful to understand the major results. 2. Line 481: typo error of "enhance".
--

REVIEWER	Chudyk, Anna University of Manitoba
REVIEW RETURNED	10-Nov-2022

GENERAL COMMENTS	You have done a great job with the revision. Congrats!!!! My only outstanding suggestion is that you provide the number of patients that received the intervention in order to clearly define your sampling (if you have this info). Right now you provide the number of patients that received the intervention and consented for further contact.
---

VERSION 2 – AUTHOR RESPONSE

Thank you very much for reviewing this manuscript. Please find responses to each point raised by the reviewer(s) below. All changes to the manuscript are highlighted with track changes (all line numbers refer to the marked copy).

1. In the first paragraph of discussion, is it possible to speak out the main points from the findings, instead of generally describe the themes titles? It will be more straightforward and helpful to understand the major results.

We have detailed the main points from the findings (Lines 473-478):

Our inductive approach to analysis found four overarching themes: i) the potential to make a difference, ii) a solution to address the gap in no man's land, iii) prioritising and making it happen and iv) personalised information and supportive conversations for taking on the advice. These themes provide an overarching message of acceptability for the LEGS intervention by highlighting benefits at the individual and system-level and by outlining strategies for increasing impact and uptake.

2. Line 481: typo error of "enhance".

Many thanks for bringing this to our attention. We have corrected the typo error (Line 483).

3. My only outstanding suggestion is that you provide the number of patients that received the intervention in order to clearly define your sampling (if you have this info).

We have added the number of patients that received the intervention (Lines 158-160):

In total, 106 patients received the intervention. All patients who had received the intervention and who gave consent for the researcher to make contact about the qualitative study were approached for participation (n=26).

We hope that these responses and improvements make the paper suitable for publication in BMJ Open.

Yours faithfully,

Clair Le Boutillier